# Six-fold increase of atmospheric $pCO_2$ during the Permian–Triassic mass extinction

Yuyang Wu [1,2], Daoliang Chu [1 ✉], Jinnan Tong[1], Haijun Song [1], Jacopo Dal Corso [1], Paul B. Wignall[3], Huyue Song[1], Yong Du[1] & Ying Cui[2 ✉]

The Permian–Triassic mass extinction was marked by a massive release of carbon into the ocean-atmosphere system, evidenced by a sharp negative carbon isotope excursion. Large carbon emissions would have increased atmospheric $pCO_2$ and caused global warming. However, the magnitude of $pCO_2$ changes during the PTME has not yet been estimated. Here, we present a continuous $pCO_2$ record across the PTME reconstructed from high-resolution $\delta^{13}C$ of $C_3$ plants from southwestern China. We show that $pCO_2$ increased from 426 +133/−96 ppmv in the latest Permian to 2507 +4764/−1193 ppmv at the PTME within about 75 kyr, and that the reconstructed $pCO_2$ significantly correlates with sea surface temperatures. Mass balance modelling suggests that volcanic $CO_2$ is probably not the only trigger of the carbon cycle perturbation, and that large quantities of $^{13}C$-depleted carbon emission from organic matter and methane were likely required during complex interactions with the Siberian Traps volcanism.

[1] State Key Laboratory of Biogeology and Environmental Geology, School of Earth Sciences, China University of Geosciences, Wuhan, China. [2] Department of Earth and Environmental Studies, Montclair State University, Montclair, NJ, USA. [3] School of Earth and Environment, University of Leeds, Leeds, UK. ✉email: chudl@cug.edu.cn; cuiy@montclair.edu

The Permian–Triassic mass extinction (PTME; ca. 252 Ma) coincided with rapid global warming that produced one of the hottest intervals of the Phanerozoic[1–5], which was likely triggered by a massive release of greenhouse gases[6,7]. The emplacement of the Siberian Traps large igneous province has been widely suggested as the ultimate trigger for the extinction of ~90% of marine species and ~70% of terrestrial vertebrate species at the Permian–Triassic boundary[8], with major losses amongst plants (e.g. refs. [9,10]). Alongside volcanic degassing, $CO_2$, $SO_2$, and halogen volatiles were likely released due to thermal metamorphism by Siberian Traps' intrusions into organic-rich sediments[6,7,11]. The global negative carbon isotope excursion (CIE) found in both marine and terrestrial settings at the PTME (for a review, ref. [12]) indicates a major carbon cycle perturbation in the ocean-atmosphere system, which implies a rise in the atmospheric $CO_2$ levels ($pCO_2$). However, $pCO_2$ changes during the PTME still remain poorly constrained.

On the one hand, records of $pCO_2$ from proxies (stomata index, palaeosol carbonates, and biomarkers) are mainly focused on the late Permian and/or Phanerozoic long-term trends without detailed $pCO_2$ data for the earliest Triassic (refs. [13–17]). On the other hand, various models show large variability of peak $pCO_2$ estimates, because of the different assumed background $pCO_2$ levels (e.g. refs. [18–20]). Hence, there is a pressing need for a continuous proxy-based and high-resolution record of $pCO_2$ during the PTME. Understanding the magnitude of $pCO_2$ changes during past hyperthermals is indeed crucial to understand the possible imminent environmental effects of today's $CO_2$ increase: $pCO_2$ has risen from 280 to more than 400 ppmv in the last 150 years and is projected to go higher[21].

Experiments on living $C_3$ plants (in the field and in growth chambers) suggest that carbon isotope fractionation ($\Delta^{13}C$) during photosynthesis increases with increasing $CO_2$ levels, lowering the carbon isotope signature of $C_3$ plants ($\delta^{13}C_p$)[22]. Based on this relationship, $\Delta^{13}C$ calculated from $\delta^{13}C_p$ measured in fossil $C_3$ plants remains can be used as a proxy for past $pCO_2$[23]. This proxy successfully reproduced ice-core records of $pCO_2$ for the Last Glacial Maximum[23], and has been applied to reconstruct

$pCO_2$ during Early Eocene hyperthermals[24], the Cretaceous Period[25], and the Toarcian Oceanic Anoxic Event[26].

Here, we present high-resolution $\delta^{13}C$ records of fossil $C_3$ plant remains from sedimentary successions of southwestern China. Using the $\delta^{13}C$ data of $C_3$ plants, we calculated a six-fold increase of atmospheric $pCO_2$ during the PTME, from 426 +133/−96 ppmv to 2507 +4764/−1193 ppmv. Furthermore, the $pCO_2$ estimates are compared with carbon isotope mass balance calculations showing that in addition to volcanic $CO_2$, large quantities of $^{13}C$-depleted carbon emission from organic matter and methane were likely required to trigger the observed global negative CIE in the exogenic carbon pool.

## Results and discussion

**High-resolution terrestrial carbon isotope records.** We present high-resolution terrestrial organic carbon isotope records ($\delta^{13}C_{org}$) from plant cuticles, wood and bulk organic matter (OM) together with our previous work[10] from four terrestrial Permian–Triassic boundary sections (Chahe, Jiucaichong, core ZK4703 and Chinahe) in southwestern China (Supplementary Fig. 1; Supplementary Fig. 2). The $\delta^{13}C$ of bulk OM and $C_3$ plant remains from the four study sections exhibit nearly identical secular trends (Fig. 1). Each profile can be divided into four stages: (1) a pre-CIE stage, (2) an onset of the negative CIE (onset of CIE) stage, (3) a prolonged CIE body stage and (4) a post-CIE stage. In the pre-CIE stage, $\delta^{13}C_{org}$ records from the Xuanwei Formation are characterized by steady values around −25.0‰ (Fig. 1). The synchronous, prominent onset of CIEs with peak values of −32‰ occurs at the bottom of the Kayitou Formation. Subsequently, the onset of the CIE stage is followed by a prolonged interval with sustained low values (ca. −30‰) through the whole Kayitou Formation, interrupted by a slight positive shift immediately after the onset of CIE. A recovery to slightly higher $\delta^{13}C_{org}$ values (−28‰ to −26‰) starts in the uppermost part of the Kayitou Formation and the Dongchuan Formation. Previously published terrestrial $\delta^{13}C_{org}$ profiles in southwestern China (e.g. refs. [27,28]) all belong to mixed organic carbon source

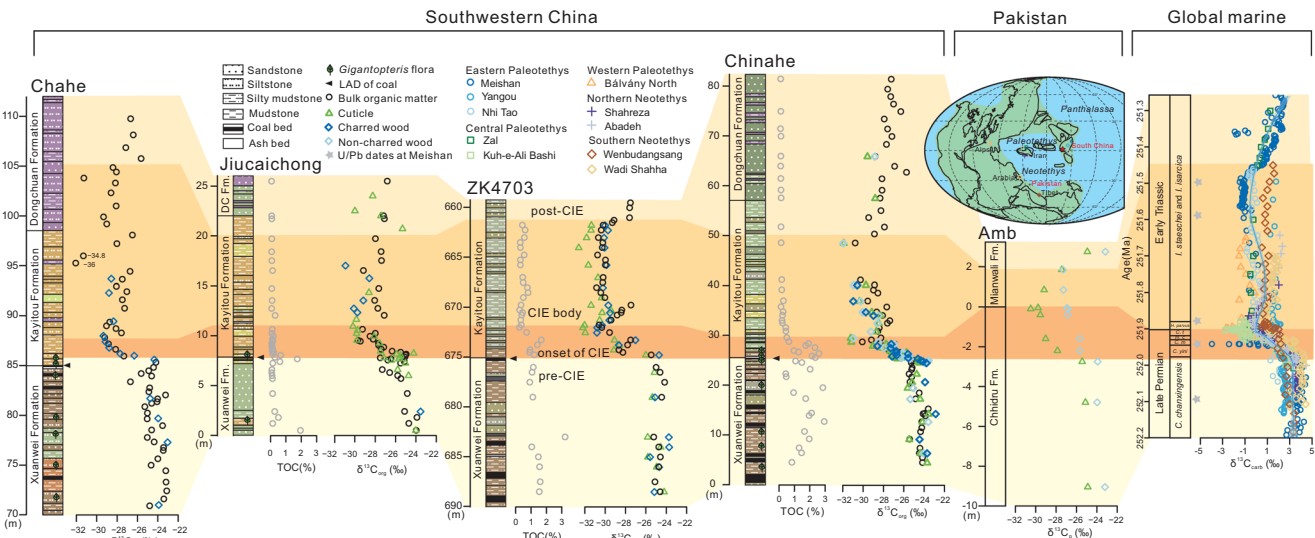

**Fig. 1 Carbon isotope excursion trend recorded in global terrestrial C₃ plants and marine carbonates.** The secular carbon isotope excursion (CIE) trend can be divided into four stages (i.e. pre-CIE, onset of CIE, CIE body and post-CIE) in terrestrial bulk organic matter, C₃ plants and marine carbonates, and are shown as different color fields. The last appearance datum (LAD) of coal beds and *Gigantopteris* flora distributions represent the coal gap and collapse of tropical peatlands respectively[10,45]. Carbon isotope (δ13C) data source: Chahe (δ13C of bulk organic matter from ref. [27]; δ13C of plants data from this study), Jiucaichong (this study), ZK4703 core and Chinahe (δ13C data in this study together with our previous work[10]), Amb (Pakistan)[32] and global marine carbonate δ13C (Methods). The locations of marine and terrestrial carbon isotope profiles are shown in the late Permian palaeogeographic map.

in bulk OM. Few unusually negative values (< −34‰) observed in the upper Kayitou Formation, e.g., in a published record from Chahe[27], are statistical outliers and local signals, as such negative values are not observed in our high-resolution study. These outliers may be caused by local $^{13}C$-depleted samples possibly containing an algal and/or bacterial component[29].

The four-stage terrestrial $\delta^{13}C_{org}$ trend is also seen in the marine carbonate carbon isotope ($\delta^{13}C_{carb}$) records (Fig. 1). A total of 10 global-distributed marine Permian–Triassic boundary sections with both high-resolution $\delta^{13}C_{carb}$ and conodont biostratigraphy were integrated as a global marine $\delta^{13}C_{carb}$ profile, using the age model from the Meishan Global Stratotype Section and Point (GSSP)[30] (Supplementary Fig. 3; Supplementary Fig. 4). These newly compiled global $\delta^{13}C_{carb}$ records are nearly identical to those published previously (e.g. ref. [12]).

**$pCO_2$ estimates based on $\Delta^{13}C$ of fossil plants.** Constraining the magnitude of the CIE is crucial to estimate accurate mass, rate, and source of the $^{13}C$-depleted carbon released during the PTME[20]. CIE magnitudes show large variations between different localities and substrates because they can be affected by multiple factors[31]. $\delta^{13}C_p$ profiles from southwestern China (low latitude) and Pakistan[32] (middle latitude) exhibit CIE magnitudes of ca. −7‰ and ca. −5.5‰ respectively, which are significantly larger than the ca. −3.5‰ marine CIE magnitude estimated from global marine $\delta^{13}C_{carb}$ records (Fig. 1). Data compilation confirms this discrepancy: terrestrial CIE magnitudes range from −3.6‰ to −6.1‰ (bulk OM, 25th percentile to 75th percentile, $n = 29$), and from −5.2‰ to −7.1‰ ($C_3$ plants, $n = 9$), whereas marine CIE magnitudes range from −3.0‰ to −4.7‰ ($n = 69$) (Fig. 2; Supplementary Table 1). As shown both in modern and fossil plants, elevated atmospheric $pCO_2$ was likely responsible for amplifying the magnitude of the CIE in the terrestrial $\delta^{13}C_p$ record due to an increase of $\Delta^{13}C$[22,33]. Therefore, following the relationship between $\Delta^{13}C$ and $pCO_2$ developed by Cui and Schubert[24] (Methods), we could calculate the $pCO_2$ across the PTME. The $\Delta^{13}C$ was calculated using the $\delta^{13}C_p$ records of four study sections from southwestern China, and the $\delta^{13}C_{CO2}$ (the $\delta^{13}C$ of atmospheric $CO_2$; Supplementary Fig. 5) calculated from the global marine $\delta^{13}C_{carb}$ compiled in this study. As explained above, this is possible because the marine and terrestrial $\delta^{13}C$ records are closely comparable and can be readily correlated (Fig. 1), the correlation being supported also by biostratigraphy (flora and conchostracans), and radioisotope dating (Supplementary Fig. 6; Supplementary Information). The initial, background late Permian $pCO_{2(t = 0)}$ is set in our calculations at 425 ± 68 ppmv based on the late Changhsingian $pCO_2$ estimates calculated by Li et al.[16] using stomatal ratio method and mechanistic gas exchange model for fossil conifers from the Dalong Formation in South China, with good age control and reliable taxonomy.

Our estimates (Fig. 3) show that $pCO_2$ was moderately low (426 +133/−96 ppmv) at 252.1 Ma within the pre-CIE stage (upper part of conodont *Clarkina changxingensis* zone). Subsequently, the $pCO_2$ began to increase rapidly in the *Clarkina yini* zone, reaching a maximum level (2507 +4764/−1193 ppmv), immediately after the Permian–Triassic boundary (*Hindeodus parvus* zone). This near six-fold increase of atmospheric $pCO_2$ occurred within ~75 kyr and coincided with the onset of the global CIE. The $pCO_2$ remained high (ca. 1500 to 2500 ppmv) immediately after the onset of the CIE, with only one transient drop (down to ca. 1300 ppmv). Coupled to the recovery of $\delta^{13}C$, $pCO_2$ drops to ca. 700 ppmv at the top of *Isarcicella isarcica* zone. Atmospheric $CO_2$ levels show a close coupling with estimated sea surface temperatures ($r = +0.60$, $p < 0.001$, $n = 173$; Supplementary Fig. 7), implying that $CO_2$ was likely the dominant greenhouse gas across the PTME, although the

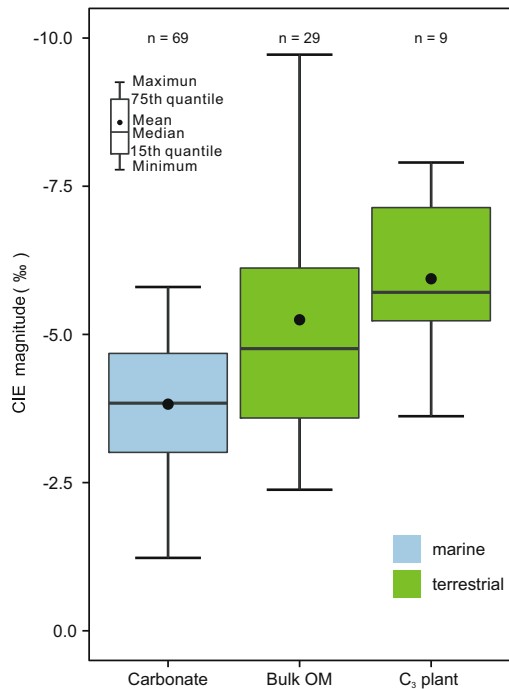

**Fig. 2 Boxplot of carbon isotope excursion magnitudes for three substrates.** Carbon isotope excursion (CIE) magnitudes of marine carbonate, terrestrial bulk organic matter (OM), and terrestrial $C_3$ plant compiled from the literature and this study. The magnitude of the terrestrial CIE is larger compared to the marine CIE magnitude. The Wilcoxon test suggests that the CIE magnitude between marine and terrestrial substrate is statistically different (Supplementary Table 1, p < 0.001). A Kruskal-Wallis test further shows the significant difference of CIE among marine carbonate, terrestrial bulk organic matter and $C_3$ plant groups ($p < 0.001$). The "$n$" value represents the number of $\delta^{13}C$ profiles.

contribution of other greenhouse gases such as methane and water vapor cannot be excluded here. The six-fold increase of atmospheric $pCO_2$, together with a 10 °C increase in sea surface temperatures estimated from low latitude conodont oxygen isotope (Fig. 3) implies Earth system sensitivity (ESS) of 3.9 °C per doubling of $CO_2$ if we assume ESS equals to $\Delta T/\log_2[pCO_{2(peak)}/pCO_{2(background)}]$[34]. This is consistent with a previous estimate of the Permian–Triassic ESS[35] and the IPCC equilibrium climate sensitivity range of 1.5 to 4.5 with a median of 3.0[36], suggesting slow feedbacks operated in the geologic past. However, climate model simulations reveal that the increase of SST in high latitude should be higher than low latitude[37]. As a result, the 10 °C SST increase in low latitude might underestimate the global SST increase, which leads to an underestimate of the Earth system sensitivity during the PTME.

**Comparison with previous studies and uncertainty.** Previous $pCO_2$ estimates around the Permian–Triassic boundary (Fig. 3; Supplementary Table 2) come from stomatal proxies[16,17], palaeosol carbonates[13,14], phytane[15] and carbon cycle modelling (e.g. refs. [18–20]). Published proxy-based $pCO_2$ reconstructions are mostly for the late Permian, within long-term and very low-resolution Phanerozoic records. Stomata-based estimates from modified fossil *Ginkgo* stomatal index method[17] gave $pCO_2$ around 400–800 ppmv in the latest Permian, but with poor age constraint and high taxonomic uncertainty[16]. Latest Permian $pCO_2$ from $\delta^{13}C$ of palaeosol carbonates from Texas, US[13], was calculated at 400 ppmv[38,39] (re-calculated by ref. [38] correcting the assumed soil respired $CO_2$ concentration), but latest Permian palaeosol carbonate record from the Karoo Basin[14] shows higher

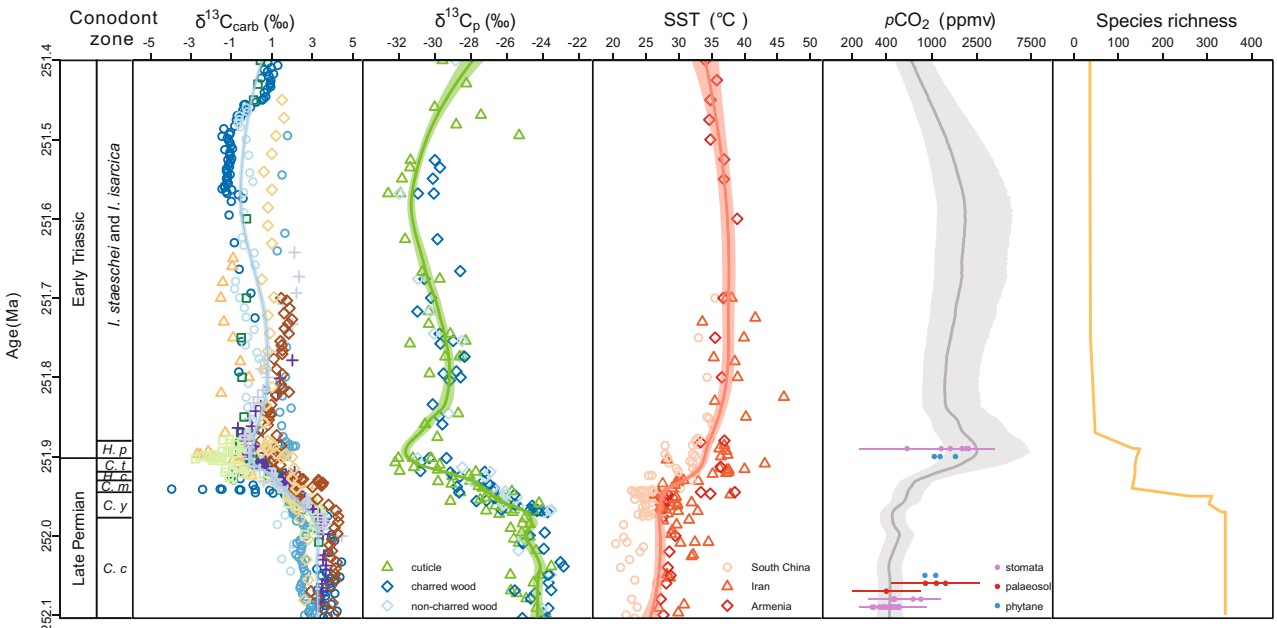

**Fig. 3 Summary of Permian–Triassic boundary proxy data and reconstructed *p*CO₂ changes.** The radiometric ages are from the Meishan section[30]. Conodont zones are those of the Meishan section. Global marine carbonate carbon isotope (δ¹³Ccarb) compiled from ten sections (Methods). Land C₃ plant carbon isotopes profile (δ¹³Cp) is from the four study sections in southwestern China (Fig. 1). Sea surface temperature (SST) was calculated based on conodont δ¹⁸O values from South China (Meishan and Shangsi)[1–3], Iran (Kuh-e-Ali Bashi and Zal)[4] and Armenia (Chanakchi)[5]. The blue, green and red lines represent the LOESS fit curve for δ¹³Ccarb, δ¹³Cp and SST, respectively, while light blue, green, and red shaded area represent 68% confidence intervals (standard errors calculated from LOESS). Reconstruction of atmospheric *p*CO₂ based on carbon isotope fractionation in C₃ land plant (on a log scale). Median values of the 10,000 re-samplings determined by Monte Carlo uncertainty propagation are shown as dark gray line. The 68% confidence intervals for *p*CO₂ are showed as light gray shaded area (lower limit and upper limit represent the 16th and 84th percentiles respectively). Previous reported *p*CO₂ estimates based on stomata[16,17], palaeosol carbonates[13,14] and phytane[15] are shown as points with error bar (Supplementary Table 2). Marine species richness data show the two pulse mass extinction[8]. I.——Isarcicella; C. c——Clarkina changxingensis; C. y——Clarkina yini; C. m——Clarkina meishanensis; H. c——Hindeodus changxingensis; C. t——Clarkina taylorae; H. p——Hindeodus parvus.

values (883–1325 ppmv). Similarly, the δ¹³Cphytane-based *p*CO₂ estimates show that CO₂ levels during Changhsingian could have ranged from 873 to 1085 ppmv[15]. The few earliest Triassic peak *p*CO₂ estimates from stomatal[17] and phytane[15] proxies show significant variation (600–2100 ppmv). Simulations with various climate models (e.g. carbon cycle box modelling[18,19] and cGENIE[20]) show major variability of peak *p*CO₂ values (1000–9380 ppmv; Supplementary Table 3), using a large range of assumed background *p*CO₂.

Several effects, especially diagenesis[40], chemical treatment[41], plant taxonomic changes[42] and precipitation[43,44] can influence the δ¹³Cp and consequently affect *p*CO₂ estimates. The original signals of δ¹³Cp values can potentially be altered by diagenesis during burial[40] and chemical treatment during sample preparation[41]. However, the method we use to calculate palaeo-*p*CO₂ considers a relative change of the Δ¹³C that minimizes these biases (Methods). A dramatic plants turnover occurred in southwestern China during the PTME, with a *Gigantopteris* flora (spore plant) in the Xuanwei and basal Kayitou formations replaced by an *Tomiostrobus* (spore plant) and *Peltaspermum* (seed plant) dominated flora[45]. Experiments on modern plants indicate lower Δ¹³C in seed plants than in spore plants[42]. Using a plant assemblage including a mixture of different taxa and plant tissues is better than using single species and plant remains when using δ¹³Cp as *p*CO₂ proxy[33]. In this study we used a mixture of different plant tissues (i.e. cuticle, charred wood and non-charred wood), which very likely includes different plant taxa.

An increase of the mean annual precipitation (MAP) can also increase Δ¹³C[44,46]. This effect is negligible in sites experiencing

high precipitation (>1500 mm/yr)[47], such as the studied area in southwestern China, which was a humid, equatorial peatland during the PTME[45]. The plant community changed from *Gigantopteris* flora-dominated rainforest ecosystem to isoetalean-dominated (lycophyte) herbaceous vegetation that inhabited the surrounding margins of coastal oligotrophic lakes, which indicate fairly constant precipitation regimes during the PTME interval[48,49]. The sedimentology of the Xuanwei and Kayitou formations suggests there was no significant precipitation change across the mass extinction (Supplementary Fig. 8; ref. [50]). In contrast, low MAP can explain the smaller magnitude of the CIE (<3%) recorded at the PTME in the semi-arid locations of Karoo Basin and North China[31]. In summary, the persistently humid condition in southwestern China was unlikely to have affected plant Δ¹³C, thus the *p*CO₂ estimates are considered robust. A Monte Carlo method has been applied to evaluate the uncertainties (Supplementary Information; Supplementary Fig. 9), which reveals that the uncertainty in the *p*CO₂ increases with increasing *p*CO₂, as seen in the previous studies[51].

**Potential source of ¹³C-depleted carbon during the PTME.** The ultimate source of ¹³C-depleted carbon capable to trigger the observed negative CIE, is widely debated. Several climate models of varying complexities (e.g. simple box models[18,19] and cGENIE[20]) use different light carbon sources to fit the δ¹³C of marine carbonates (Supplementary Table 3). Proposed ¹³C-depleted carbon sources include biotic or thermogenic methane (δ¹³C ≈ −60% to −40%; e.g. ref. [18]), CO₂ from thermal metamorphism or rapid oxidation of organic-rich rock (δ¹³C ≈ −25%; e.g. ref. [6,19,52,53]), and

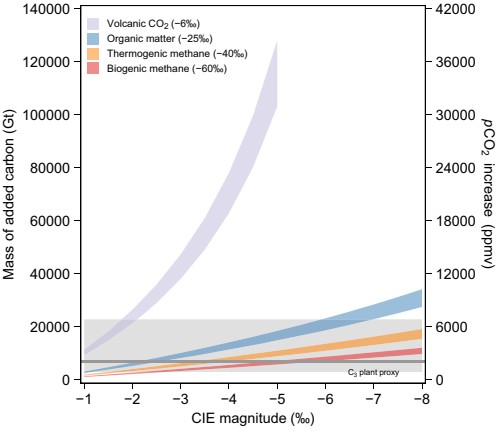

**Fig. 4 Mass of added carbon estimated from carbon isotope mass balance calculation.** Four different scenarios including volcanic $CO_2$ ($\delta^{13}C =$ −6‰), organic matter ($\delta^{13}C = −25$‰), thermogenic methane ($\delta^{13}C = −40$‰) and biogenic methane hydrate ($\delta^{13}C = −60$‰). Second y-axis converts the mass of added carbon to an increase in atmospheric $p$CO$_2$ based on the earth system model (1 Gt C = 0.3 ppmv $CO_2$)[55,56]. Gray shaded area represents the 68% confidence intervals of $p$CO$_2$ increase (2081 +4764/−1193 ppmv) estimated from the $C_3$ plant proxy.

volcanic $CO_2$ ($\delta^{13}C \approx$ −6%; e.g. ref. [54]) or a combination of these sources[20]. We performed a simple carbon isotope mass balance to evaluate the most likely $^{13}C$-depleted carbon source[55,56]. Under the assumption of four possible $^{13}C$-depleted carbon sources (i.e. volcanic $CO_2$, organic matter, thermogenic methane, and biogenic methane), the mass of released carbon was calculated (Fig. 4) and compared with our $p$CO$_2$ rise estimates (2081 +4764/−1193 ppmv). Our calculated $p$CO$_2$ mostly falls within the range of model results for organic matter and methane release scenarios (Fig. 4), supporting the hypothesis that these more $^{13}C$-depleted sources than volcanic $CO_2$ are required to contribute to the global carbon cycle perturbation. There are some U-Pb dating[7] and field evidence[6,57] show that the organic-rich sediment intruded by Siberian Traps sill likely provided massive $^{13}C$-depleted $CO_2$ and thermogenic methane, which may have been the ultimate trigger of the global CIE and significant increase in atmospheric $CO_2$. However, due to the limitation of the $C_3$ plant proxy, the uncertainty of $p$CO$_2$ is significantly larger at high $CO_2$ levels (Supplementary Fig. 9). Therefore, volcanic $CO_2$ source could still have made a contribution to the global carbon cycle perturbation.

The best estimates for mass of added carbon based on a 3.5% carbonate CIE magnitude and a source with $\delta^{13}C$ of −25% to −60%, suggest that at least 3900~12,000 Gt carbon were added into the ocean-atmosphere system during the PTME. Previous estimates (15,000–20,000 Gt C) were based on an assumed *ca.* 5.5% negative shift of $C_3$ plant in simple mass balance calculations[32] and might therefore have overstated the amount of added carbon. The ca. 7% CIE in $C_3$ plants, amplified by $p$CO$_2$ increase, also produces an over-estimate in the mass balance calculation (Fig. 4). Our estimates of the amount of injected carbon are also smaller than those calculated by the cGENIE climate model (7,000~22,400 Gt C)[20], because the ~5% magnitude of $\delta^{13}C_{carb}$ from Meishan used in the calculations is too large compared to the global carbonate records. However, simple mass balance calculations don't consider global carbon cycle fundamental processes and changes through time, like carbon weathering and burial rates during the studied interval. In addition, the size of the DIC reservoir is usually assumed to be the size of the background surface carbon reservoir, because of poor understanding of atmosphere carbon reservoir size[58]. These

limitations might lead to an underestimate of the total mass of added $CO_2$.

Carbon emission caused prolonged high $p$CO$_2$ and high temperature (ca. 35 °C) during the earliest Triassic (*H. parvus* and *I. isarcica* zones) and may have lasted for > 500 kyr (Fig. 3). This lengthy phase of extreme warmth likely implies prolonged carbon emissions into ocean-atmosphere system from continued eruption of the Siberian Traps volcanism, and/or reduced carbon sequestration rate, potentially due to lower consumption of atmospheric $CO_2$ through reduced organic carbon burial and the possible failure of the silicate weathering thermostat[59].

## Methods

**Sample treatment and analysis.** In total, 68 samples from Chinahe, 41 samples from ZK4703 and 40 samples from Jiucaichong were analyzed for bulk organic carbon isotopes. Samples were crushed to fine powder (<200 mesh), and ∼2 g powder were weighed, placed into a centrifuge tube and treated with 3 mol/L HCl for 24 h to remove the carbonate. Then the treated samples were rinsed with ultrapure water repeatedly until neutralized and finally dried at 35 °C. For $C_3$ plants $\delta^{13}C$ analysis, 45 samples from Chinahe, 26 samples from ZK4703, 30 samples from Jiucaichong and 13 samples from Chahe were treated with concentrated HCl and HF, then sieved over 500 μm and a 100 μm mesh screen to get the 100~500 μm particles. $C_3$ plant fragments, including cuticle, non-charred wood and charred wood (charcoal), were picked under the microscope. The $\delta^{13}C_{org}$ analyses were performed by using an elemental analyzer (EA) coupled to an isotope ratio mass spectrometer (Thermo Delta V Advantage) at the State Key Laboratory of Biogeology and Environmental Geology of the China University of Geosciences (Wuhan). The results were calibrated using certified secondary references standards: USGS40 ($\delta^{13}C = −26.39$‰) and UREA ($\delta^{13}C = −37.32$‰) and given in per mil (%) relative to Vienna Pee Dee Belemnite (VPDB) with analytical precision better than ± 0.2%. A Multi EA 4000-analyzer was used for TOC at China University of Geosciences (Wuhan), yielding an analytical precision of 1.5%.

**Carbon isotope compilation and estimate of CIE magnitude.** In order to estimate a reliable magnitude of the CIE, the carbon isotope profiles that record a roughly complete CIE shape with pre-CIE and CIE body are selected in our study. The compilation consists of 69 marine carbonate carbon isotope ($\delta^{13}C_{carb}$) profiles and 38 terrestrial $\delta^{13}C_{org}$ profiles. The $\delta^{13}C_{carb}$ profiles recording complete negative CIE are from Eastern Palaeotethys ($n = 29$), Western Palaeotethys ($n = 19$), Central Palaeotethys ($n = 4$), Northern Neotethys ($n = 4$), Southern Neotethys ($n = 10$) and Panthalassa ($n = 3$). Bulk marine organic matter $\delta^{13}C$ records were not included, because they often represent a mix of various organic components (both marine and terrestrial). The sedimentary facies belong to a range of shallow shelf, deep shelf, and slope environments. The few reported $\delta^{13}C_{carb}$ profiles from deep basins are ignored in this compilation (e.g. Shangsi section), because the elevated water stratification and large vertical $\delta^{13}C$ DIC gradients at deep basin sites during Permian–Triassic crisis could cause large CIE magnitudes[60]. A total of 38 terrestrial $\delta^{13}C_{org}$ records are reviewed from eight terrestrial basins including western Guizhou and eastern Yunnan in southwestern China ($n = 14$), Junggar Basin ($n = 3$), Turpan Basin ($n = 1$), North China ($n = 3$), Central European Basin ($n = 1$), Bowen Basin ($n = 2$), Sydney Basin ($n = 7$), Karoo Basin ($n = 1$) and three oceanic regions where organic matter (OM) in samples are $C_3$ plants or a mix of organic matter dominated by $C_3$ plants including South China ($n = 1$), Boreal realm ($n = 2$) and South Neotethys ($n = 3$). Among these terrestrial $\delta^{13}C_{org}$ profiles, there are nine records of $\delta^{13}C$ records from $C_3$ plant (5 $\delta^{13}C_{wood}$ and 4 $\delta^{13}C_{cuticle}$) from Meishan section, Amb section in South Neotethys, southwestern China, and others are all bulk $\delta^{13}C_{org}$ profiles.

In order to demonstrate the difference of marine and terrestrial CIE magnitudes, carbon isotope values immediately before the CIE ($\delta^{13}C_{background}$) and peak values ($\delta^{13}C_{peak}$) are used to calculate the magnitude of the CIE ($\delta^{13}C_{peak}$ − $\delta^{13}C_{background}$). Note that 31 pairs of $\delta^{13}C_{background}$ and $\delta^{13}C_{peak}$ values are from marine sections that are well constrained by latest Permian conodont occurrences (e.g. *C. changxingensis*, *H. praeparvus*, *H. latidentatus* zones) and earliest Triassic conodont (*H. parvus* and *I. isarcica* zones) occurrences. To test if the discrepancy of the CIE magnitude in different substrates (marine carbonate, terrestrial bulk OM, terrestrial $C_3$ plant tissues) is statistically significant, we used a non-parameter Kruskal-Wallis test (function *kruskal.test*) using R software. The Wilcoxon (function *wilcox.test*) test was performed in R software to determine whether means of two independent groups (marine vs. terrestrial) are equal or not. Boxplots were drawn to visualize discrepancy in CIE magnitude of different substrates. All statistical analyses and graphing functions were undertaken using R.

**$C_3$ plant proxy.** The carbon isotope fractionation in $C_3$ plants ($\Delta^{13}C$) and atmospheric $p$CO$_2$ is described as a hyperbolic relationship[22–24,61]:

$$\Delta^{13}C = \frac{(A)(B)(p\text{CO}_2 + C)}{A + (B)(p\text{CO}_2 + C)} \qquad (1)$$

The original $\delta^{13}C$ signals in $C_3$ plant can be altered by several effects (e.g. diagenesis[40], chemical treatments[41]), that potentially influence $pCO_2$ calculations. In order to minimize this effect, the data set is analyzed by a relative change in the $\Delta^{13}C$ value between the time of interest ($t$) and a reference time ($t = 0$), designated as $\Delta\left(\Delta^{13}C\right)$:

$$\Delta\left(\Delta^{13}C\right) = \Delta^{13}C_{(t)} - \Delta^{13}C_{(t=0)} \qquad (2)$$

which can be expanded as:

$$\Delta\left(\Delta^{13}C\right) = \frac{(A)(B)\left(pCO_{2(t)} + C\right)}{A + (B)\left(pCO_{2(t)} + C\right)} - \frac{(A)(B)\left(pCO_{2(t=0)} + C\right)}{A + (B)\left(pCO_{2(t=0)} + C\right)} \qquad (3)$$

By rearranging Eq. (3), $pCO_{2(t)}$ at any given time can be calculated by

$$pCO_{2(t)} = \frac{\Delta(\Delta^{13}C) \cdot A^2 + \Delta(\Delta^{13}C) \cdot A \cdot B \cdot pCO_{2(t=0)} + 2 \cdot \Delta(\Delta^{13}C) \cdot A \cdot B \cdot C + \Delta(\Delta^{13}C) \cdot B^2 \cdot C \cdot pCO_{2(t=0)} + \Delta(\Delta^{13}C) \cdot B^2 \cdot C^2 + A^2 \cdot B \cdot pCO_{2(t=0)}}{A^2 \cdot B - \Delta(\Delta^{13}C) \cdot A \cdot B - \Delta(\Delta^{13}C) \cdot B^2 \cdot pCO_{2(t=0)} - \Delta(\Delta^{13}C) \cdot B^2 \cdot C} \qquad (4)$$

where $A$, $B$, $C$ are curve fitting parameters. Values for $A$ and $B$ are $28.26 \pm 0$ and $0.22 \pm 0.028$, respectively[51], which could produce more robust $pCO_2$ estimates compared with other parameter values in subsequent research[33]. The $C$ is the function of the $A$ and $B$ values $[C = A \times 4.4/((A - 4.4) \times B)]$. The $pCO_{2(t=0)}$ is equal to the $pCO_2$ level before the negative CIE, determined from independent stomatal proxies based on fossil conifers from the Dalong Formation in South China[16]. Because of the good age control (Clarkina changxingensis conodont zone), reliable taxonomy and calculation method, these stomatal estimates are considered as robust $pCO_2$ estimates before CIE. The mean value for the stomatal estimates is $425 \pm 68$ ppmv set as $pCO_{2(t=0)}$. The $\Delta^{13}C$ is the carbon isotope fractionation between atmospheric $CO_2$ and plant organic carbon ($\Delta^{13}C = \left(\delta^{13}C_{CO_2} - \delta^{13}C_p\right)/\left(1 + \delta^{13}C_p/1000\right)$). Thus, the $\Delta(\Delta^{13}C)$ can be calculated by

$$\Delta\left(\Delta^{13}C\right) = \left(\delta^{13}C_{CO_2(t)} - \delta^{13}C_{p(t)}\right)/\left(1 + \delta^{13}C_{p(t)}/1000\right) - \left(\delta^{13}C_{CO_2(t=0)} - \delta^{13}C_{p(t=0)}\right)/\left(1 + \delta^{13}C_{p(t=0)}/1000\right) \qquad (5)$$

where $\delta^{13}C_{p(t=0)}$ and $\delta^{13}C_{p(t)}$ are $\delta^{13}C$ values in $C_3$ plant at reference time ($t = 0$) and the time of interest ($t$). The values for $\delta^{13}C_{p(t=0)}$ is determined as $-24.42 \pm 0.5\%$, whose age equals to Clarkina changxingensis conodont zone that occurred slightly earlier than the onset of the CIE[16]. We suggest that a mixture of $\delta^{13}C$ in $C_3$ plant cuticle, charred wood and non-charred wood from southwestern China (without $\delta^{13}C$ of bulk OM) provides the best choice as $\delta^{13}C_{p(t)}$ input data for three reasons. Firstly, using the $\delta^{13}C$ values of $C_3$ plant tissues (e.g. cuticle, wood) can minimize the influence of varying OM sources from mixed soils and sediments. Several previous reports on terrestrial $\delta^{13}C_{org}$ in western Guizhou and eastern Yunnan, South China have recorded the negative CIE[27,28,62,63], but all the data are not $\delta^{13}C$ from $C_3$ plants and not suitable for $pCO_2$ calculation. Secondly, Eq. (1) is based on the combination of carbon isotope from stem and leaf tissues of chamber plants. Thirdly, the mixture of different micro plant tissues (i.e. cuticle, charred wood and non-charred wood) would contain different plant fossils species that is suggested to be better than a single species approach when applying this proxy[33].

The $\delta^{13}C_{CO_2(t=0)}$ and $\delta^{13}C_{CO_2(t)}$ are $\delta^{13}C$ values in atmospheric $CO_2$ at reference time ($t = 0$) and the time of interest ($t$). The temperature ($T$) dependent carbon isotope fractionation between dissolved inorganic carbon (DIC) and atmospheric $CO_2$[64] can be used to calculate $\delta^{13}C_{CO_2}$.

$$\delta^{13}C_{CO_2} = \delta^{13}C_{DIC} - (0.91 \times (-0.1141 \times T + 10.78) + 0.08 \times (-0.052 \times T + 7.22)) \qquad (6)$$

where $T$ is the sea surface temperature determined from oxygen isotopes of conodont fossils[1–5]. $\delta^{13}C_{DIC}$ can be estimated from marine $\delta^{13}C_{carb}$ ($\delta^{13}C_{DIC} = \delta^{13}C_{carb} - 1\%$), because the carbon isotope fractionation between marine carbonate ($\delta^{13}C_{carb}$) and dissolved inorganic carbon ($\delta^{13}C_{DIC}$) is constant and independent of temperature ($\sim 1\%$)[65]. Among the global CIE compilations, 10 marine, high-resolution, non-basinal $\delta^{13}C_{carb}$ profiles are well constrained by detail conodont zones including Meishan[30], Nhi Tao[66], Yangou[67] in eastern Palaeotethys; Zal[3], Kuh-e-Ali Bashi[3] in central Palaeotethys; Bálvány North[68] in western Paloetethys; Shahreza[12], Abadeh[12] in northern Neotethys; Wadi Shahha[69], Wenbudangsang[70] in Southern Neotethys. Thus, we integrated these 10 $\delta^{13}C_{carb}$ profiles together as a global marine $\delta^{13}C_{carb}$ profile combined with a U-Pb age model[30] and high-resolution conodont zones (conodont zones from Meishan are selected as standard[71,72]). LOESS curves with 0.002 Myr spacing were fitted to the integration of global marine $\delta^{13}C_{carb}$. At each 0.002 Myr time step, the probability maximum value and standard error are identified and served as $\delta^{13}C_{carb}$ input parameters in calculations. The best degree of smoothing for LOESS fitting was determined using cross-validation method in package fANCOVA. To eliminate the potential for an uneven distribution of $\delta^{13}C_{carb}$ data, we also applied a LOESS fitting based on an 80% subsample of all data. In addition, the $\delta^{13}C_{carb}$ data from Meishan ($n = 199$) is the most abundant of all the $\delta^{13}C_{carb}$ data ($n = 707$), thus, we performed a LOESS fitting based on $\delta^{13}C_{carb}$ data without Meishan data (Supplementary Fig. 4).

In order to calculate $pCO_{2(t)}$, we need to align global marine $\delta^{13}C_{carb}$ profiles and $\delta^{13}C_p$ based on same age model. The nearly same CIE curves were divided into four stages in carbonate and $C_3$ plants records to ensure the correlation between marine and terrestrial carbon isotope profiles. The age model for four sections is showed in Supplementary Information. In addition, the LOESS method with 0.002 Myr spacing was also performed in $\delta^{13}C_p$ and temperature data to get the probability maximum value and standard error at each 0.002 Myr time step. The Monte Carlo method was employed to propagate input error[51] by the propagate package in R. All the input parameters were assumed to be Gaussian distributed with mean and standard deviations listed in Supplementary Table 4. 10,000 values for each input parameters were randomly sampled to calculate 10,000 values for each $pCO_{2(t)}$. The invalid $pCO_{2(t)}$ values (i.e., $pCO_{2(t)} < 0$ or $>10^6$ ppmv) were excluded. The 16th and 84th percentiles of the remaining estimates were determined to construct the 68% confidence interval. The positive error of the reconstructed $pCO_{2(t)}$ value represents the difference between the 84th percentile value and the median, and the negative error represents the difference between the 16th percentile value and the median. The sensitivity analysis of $C_3$ plant proxy is discussed in Supplementary information and Supplementary Fig. 9.

**Carbon isotope mass balance**. This model used to evaluate the light carbon source, following mass balance equation is modified from McInerney and Wing[55]:

$$M_{added} = \frac{\left(-CIE \times M_{background}\right)}{\left(\delta^{13}C_{peak} - \delta^{13}C_{added}\right)} \qquad (7)$$

where $M_{added}$ is the mass of carbon added into atmosphere-ocean system carbon emission. The $M_{background}$ represents initial carbon reservoir size during Permian–Triassic including ocean and atmosphere carbon inventory, but dominated by the ocean reservoir. Thus, the $M_{background}$ is assumed to be the initial marine DIC reservoir size ranging from 66,000 to 82,000 Gt[58,73]. CIE represents the global magnitude of CIE controlled only by release of light carbon effect, which is set as a series values from $-1\%$ to $-8\%$. The peak $\delta^{13}C$ value ($\delta^{13}C_{peak}$) at the event is calculated by initial isotopic composition of global carbon reservoir ($\delta^{13}C_{background}$) and CIE ($\delta^{13}C_{peak} = \delta^{13}C_{background} + CIE$). The $\delta^{13}C_{background}$ is assumed to be the initial isotopic composition of DIC reservoir 2.2% that is estimated from global marine $\delta^{13}C_{carb}$ profiles (age >252.104 Ma). The $\delta^{13}C_{added}$ is the $\delta^{13}C$ value of the carbon source causing the CIE. Four kinds of carbon sources are involved including biogenic methane buried in permafrost or seafloor ($\delta^{13}C = -60\%$), thermogenic methane ($\delta^{13}C = -40\%$), thermal metamorphism or rapid oxidation of organic-rich rock ($\delta^{13}C = -25\%$) and $CO_2$ released from direct volcanic eruption ($-6\%$). Finally, the increased $pCO_2$ is estimated from $M_{added}$ (1 Gt $C = 0.3$ ppmv; ref. [56]), and compared with reconstructed atmospheric $CO_2$ levels from $C_3$ plant proxy.

## Data availability
The authors declare that all data supporting the findings of this study are available within the paper and its supplementary file.

## Code availability
R code to run the model is available from D.L. Chu on request.

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

## Acknowledgements

Many thanks to Linhao Fang for the help of sample treatment method. We thank Wenchao Shu and Yao Wang for sample treatment. This study was supported by the National Natural Science Foundation of China (42030513, 41821001, 41530104, 42072025, 41888101), the US National Science Foundation grant no. EAR-1603051 and EAR-2026877, and also benefited from Natural Environment Research Council (UK) grant, 'Ecosystem resilience and recovery from the Permo-Triassic crisis' (grant NE/P013724/1), which is a part of the Biosphere Evolution, Transitions and Resilience (BETR) Program. This is Center for Computational & Modeling Geosciences (BGEG) publication number 2.

## Author contributions

D.L.C., Y.C., and Y.Y.W. designed this study with in-depth inputs from H.J.S., P.B.W., J.D.C., and J.N.T. Y.Y.W. and D.L.C completed the data preparation and analysis with the help from H.Y.S., Y.D., and J.D.C. Y.Y.W. and Y.C. performed the calculations. Y.Y.W and D.L.C wrote the main text and supplementary materials with inputs from all authors.

## Competing interests

The authors declare no competing interests.
