## [Peer Review File · Nature Communications]

REVIEWER COMMENTS

Reviewer #1 (Remarks to the Author):

Review of NCOMMS-20-17959-T by Y.Y. Wu and others

I have read the article by Wu and others and found it to be well-written, clearly illustrated, and fully supported by supplementary documents explaining the methodologies used and analyses performed. The paper conveys a significant new research finding that will be of broad interest to the deep time Earth history research community, and which also has modern societal implications. It is my view that the paper articulates an important, original scientific finding that should be published in Nature Communications. This finding (six-fold increase from present-day levels in atmospheric pCO₂ at the end-Permian) will inform future attempts to model and constrain palaeoenvironmental conditions across the worst biotic crisis of the Phanerozoic (so far). I will acknowledge that I am a sedimentologist and have only a basic understanding of the sedimentary geochemistry concepts and methods used in this paper. I can say that the outcomes documented in the paper are entirely consistent with findings of my own group working in eastern Australia. I appreciate the manner in which the authors have collated all available carbon isotope data and applied quality control criteria in order to rank their usefulness in evaluating end-Permian palaeoenvironmental change. We have found similar variability between sections and between geographic regions, and we have struggled to find a way of normalizing this variability. One aspect, though tangential to the point of the paper, is the lack of detail on lithologies and their sedimentology. We are given only a cursory reference to Wignall et al. (2020, in *Global and Planetary Change*), and even the Geological Setting section of the Supplementary document contains no details of the sedimentology. This may matter, because the authors assert "The sedimentology of the Xuanwei and Kayitou formations suggests there was no significant precipitation change across the mass extinction⁴⁵" (lines 155-156), yet I can find no way of assessing that statement, and having reread Wignall et al., cannot see any basis for it. I think it would assist in the documentation of the results if they were in some way tied to graphic logs of one or more of the measured sections, with details of the facies analysis also provided. The generalised lithologies included in Figure 1 do not have such information.

Chris Fielding, Lincoln, NE, USA, June 1st, 2020

Reviewer #2 (Remarks to the Author):

This paper presents atmospheric CO₂ concentrations across the Permian-Triassic boundary determined using the C₃ plant proxy. The paper is well written and well organized and the topic is of broad interest. The authors argue for a large increase in CO₂ at the boundary, but not large enough to be explained by volcanic CO₂ outgassing. The traditional explanation for this event has been Siberian Traps volcanism so this is a significant conclusion. However, I do not think that the authors can actually exclude volcanic CO₂ as a carbon source with their data. First, they use a 68% confidence interval, which is not statistically rigorous. Second, the proxy saturates at high CO₂, making it very difficult to confidently exclude very high CO₂, especially given the large uncertainty in the δ¹³C values of atmospheric CO₂ stemming from large variance in marine carbonate δ¹³C values. Therefore, the most novel contribution of the paper is questionable.

Detailed comments:

The δ¹³C value of plants is controlled by many factors. Importantly, water stress is a factor. The authors argue for humid conditions in the study region, which is a good start. What about the individual plants studied? Are they intolerant of aridity? Discussing the taxonomy and what this tells about the environments the wood comes from will be important to make a commencing argument here.

The authors have not accounted for carbon isotope change to wood that occurs during diagenesis. This will affect the results. See this recent paper:
Lukens, W. E., Eze, P., & Schubert, B. A. (2019). The effect of diagenesis on carbon isotope values of fossil wood. *Geology*, 47(10), 987-991.

There is a large spread in marine carbonate $\delta^{13}\text{C}$ values especially at the peak of the excursion where there is a range of 6‰. It is hard for me to believe that this degree of uncertainty was fully considered in the error propagation - the CO_2 range shown in Fig 3 seems too narrow. How did you consider the substantial error here? I see the confidence interval on the LOESS fit but it looks way too small, even for a 68% CI. Some of the marine carbonate records show a large excursion (-6‰) whereas others are much smaller. Given that the variable magnitude of marine CIE during the PETM has been attributed to dissolution, would you say that larger marine CIEs give a more accurate representation? The average of the marine CIE may not be the best way to proceed. Please discuss.

How does Fig. 4 support the conclusion that the CIE was not the result of volcanic CO_2 ? I see the black cross overlapping all emission scenarios. Please also discuss whether this calculations assumes rapid emission in which all the carbon goes into the atmosphere, or whether it considers mixing into the ocean. Peak CO_2 will be very different if the release rate is slow and mixing with the ocean occurs.

Lines 119-120. This is probably right, but to be scientific about it, considering the feedbacks in the carbon cycle, the correlation doesn't mean CO_2 was the control. For instance, methane is highly correlated with CO_2 and temperature during Pleistocene but we don't think of methane as being the primary control.

Line 121- where does the 10 degrees C come from? Need global mean T for this calculation.

Line 178-179 - I am not convinced that you can rule out $\text{CO}_2 > 10,000$ ppm using the data presented here. The proxy saturates at high CO_2 . Is it really possible to confidently distinguish 5000 from 10,000 ppm?

Reviewer #3 (Remarks to the Author):

Overall I find this to be a strong manuscript which will be of much interest to a broad audience. The new pCO_2 record represents one of the most high resolution records for the P-T interval. The proxy based method used to generate the new pCO_2 record is not without its critics however there is now ample experimental data which highlight the promise and strengths (and also limitation) of the new $\delta^{13}\text{C}$ isotope based proxy. The application of the proxy in this instance is appropriate and robust.

I find the CIE attribution analysis less convincing because of the range of possible combinations of both CO_2 estimates and $\delta^{13}\text{C}$ values for each time interval/ section / possible source. I recommend that the authors down play the certainty of their attribution a little more.

I have two minor issues with the CO_2 record as presented.

(1) The authors argue that a mixed sample of cuticle, charcoal and various unknown taxonomic groups decrease the likelihood that an evolutionary or phylogenetic signal in $\delta^{13}\text{C}$ will interfere or bias the CO_2 record. This is incorrect. An unknown mix of different taxonomic groups means that the potential phylogenetic bias cannot be quantitatively assessed. I agree that an assemblage approach is better than a single species approach (Porter et al) however the authors must add the

caveat that any phylogenetic bias in the CO₂ record still remains to be addressed with further study. (lines 487-488 149-150)

(2) The authors do not consider the influence of different chemical treatment on the fidelity of the original ¹³C C₃ plant signal. Baral et al (2015) showed that HF and HCL treated fossil plant material can alter the original ¹³C signal. I note in the methods that different section and different plant preservation types were processed in different ways - some with only HCL , others with both HCL and HF. Were some cuticles also picked without any chemical post treatment prior to ¹³C analysis? I request that the authors address this potential bias in their CO₂ data set. I do not see this as insurmountable. Perhaps it is a consistent bias that can be accounted for? However if different parts of the sections were processed in different ways then this inconsistent bias in the perceived 'raw ¹³C data' and how it may have effected the final CO₂ record must be evaluated. Barral A, Lécuyer C, Gomez B, Fourel F, Daviero-Gomez V. Effects of chemical preparation protocols on $\delta^{13}\text{C}$ values of plant fossil samples. *Palaeogeography, Palaeoclimatology, Palaeoecology*. 2015 Nov 15;438:267-76.

We will address reviewer's concerns pertaining to our manuscript (NCOMMS-20-17959-T), in the following paragraphs. Several of the reviewers suggested changes in wording to the text which we have included in the revised version. Those changes appear as highlights in yellow. To facilitate their review, I have included all text sent to us along with our responses.

REVIEWER COMMENTS

Reviewer #1 (Remarks to the Author):

Review of NCOMMS-20-17959-T by Y.Y. Wu and others

I have read the article by Wu and others and found it to be well-written, clearly illustrated, and fully supported by supplementary documents explaining the methodologies used and analyses performed. The paper conveys a significant new research finding that will be of broad interest to the deep time Earth history research community, and which also has modern societal implications. It is my view that the paper articulates an important, original scientific finding that should be published in Nature Communications. This finding (six-fold increase from present-day levels in atmospheric pCO₂ at the end-Permian) will inform future attempts to model and constrain palaeoenvironmental conditions across the worst biotic crisis of the Phanerozoic (so far).

Thanks for your very positive comments.

I will acknowledge that I am a sedimentologist and have only a basic understanding of the sedimentary geochemistry concepts and methods used in this paper. I can say that the outcomes documented in the paper are entirely consistent with findings of my own group working in eastern Australia. I appreciate the manner in which the authors have collated all available carbon isotope data and applied quality control criteria in order to rank their usefulness in evaluating end-Permian palaeoenvironmental change. We have found similar variability between sections and between geographic regions, and we have struggled to find a way of normalizing this variability. One aspect, though tangential to the point of the paper, is the lack of detail on lithologies and their sedimentology. We are given only a cursory reference to Wignall et al. (2020, in Global and Planetary Change), and even the Geological Setting section of the Supplementary document contains no details of the sedimentology. This may matter, because the authors assert "The sedimentology of the Xuanwei and Kayitou formations suggests there was no significant precipitation change across the mass extinction⁴⁵" (lines 155-156), yet I can find no way of assessing that statement, and having reread Wignall et al., cannot see any basis for it. I think it would assist in the documentation of the results if they were in some way tied to graphic logs of one or more of the measured sections, with details of the facies analysis also provided. The generalised lithologies included in Figure 1 do not have such information.

Chris Fielding, Lincoln, NE, USA, June 1st, 2020

Thanks for these very useful suggestions. As you mentioned, we noticed that you and your group have done detailed and novel work on the continental Permian-Triassic mass extinction in the Sydney Basin, Australia, showing a similar record to that in South China (e.g., Fielding et al., 2019 Nature Communications). According to your suggestion, we had added more details on the

sedimentology of the studied sections in the main text and Supplementary document (Supplementary Figure 7). The formation boundary was examined in detail at the outcrop sections and the lithofacies were found to be identical. Five lithofacies occur in the boundary interval, which indicated low-energy coastal swamp conditions during the Permian-Triassic boundary interval. Absence of wave processes points to a sheltered setting, either behind protective islands or on a broad, gentle shelf in which basinal processes were dampened. Superimposed on this low-energy deposition, the introduction of sand either occurred in regular, minor influxes or as major high-energy events. Such event beds indicate major flood events regularly swept through the coastal environments. In summary, no significant change in depositional style was noted between the Xuanwei and Kayitou formations, which suggested no significant precipitation change. There is no evidence for immediate aridification across the mass extinction, despite the sudden loss of coal and Permian flora observed in various sections.

Reviewer #2 (Remarks to the Author):

This paper presents atmospheric CO₂ concentrations across the Permian-Triassic boundary determined using the C3 plant proxy. The paper is well written and well organized and the topic is of broad interest. The authors argue for a large increase in CO₂ at the boundary, but not large enough to be explained by volcanic CO₂ outgassing. The traditional explanation for this event has been Siberian Traps volcanism so this is a significant conclusion.

Thanks for your revision and constructive comments.

However, I do not think that the authors can actually exclude volcanic CO₂ as a carbon source with their data. First, they use a 68% confidence interval, which is not statistically rigorous. Second, the proxy saturates at high CO₂, making it very difficult to confidently exclude very high CO₂, especially given the large uncertainty in the $\delta^{13}\text{C}$ values of atmospheric CO₂ stemming from large variance in marine carbonate $\delta^{13}\text{C}$ values. Therefore, the most novel contribution of the paper is questionable.

Thanks for your constructive comments.

For $p\text{CO}_2$ reconstruction, we think that 68% confidence (1 sd) interval used in error propagate is more statistically rigorous than 98% confidence (2 sd). As some invalid $p\text{CO}_{2(t)}$ values (i.e., $p\text{CO}_{2(t)} < 0$ or $>10^6$ ppmv) would be generated when estimating CO₂, because of the hyperbolic characteristic of this CO₂ model. We agree that the current record shows a large variance in marine carbonate $\delta^{13}\text{C}$ values, which could lead to the large uncertainty in the $\delta^{13}\text{C}$ values of atmospheric CO₂. The global carbon isotope compilation is performed to constrain the CIE magnitude. The values for CIE magnitudes that are too small or too large could not represent the global signal because of various local effects (i.e., sedimentary facies, diagenesis). As for the evaluation of the depleted carbon source, we agree with Reviewer #2 that a high uncertainty occurs under high CO₂ level. We have therefore simplified the method and used carbon isotope mass balance to evaluate the carbon source. Detailed responses to these comments are given below.

Detailed comments:

The $\delta^{13}\text{C}$ value of plants is controlled by many factors. Importantly, water stress is a factor. The authors argue for humid conditions in the study region, which is a good start. What about the individual plants studied? Are they intolerant of aridity? Discussing the taxonomy and what this tells about the environments the wood comes from will be important to make a compelling argument here.

We have added information about the plants that supports the high precipitation (Yu et al., 2015; Feng et al., 2020):

“The plant community changed from a *Gigantopteris* flora-dominated rainforest ecosystem to diminutive herbaceous vegetation that inhabited the same humid swamps or the surrounding margins of coastal lakes, which indicate fairly constant precipitation regimes during PTME interval”.

In addition, as the Reviewer #1 suggested, we had added more details on the sedimentology of the studied succession in the main text and Supplementary materials (Supplementary Fig. 7). In summary, we clarified that no significant change in depositional style between the Xuanwei and Kayitou formations is observed, suggesting no significant change in the precipitation regime.

The authors have not accounted for carbon isotope change to wood that occurs during diagenesis. This will affect the results. See this recent paper:

Lukens, W. E., Eze, P., & Schubert, B. A. (2019). The effect of diagenesis on carbon isotope values of fossil wood. *Geology*, 47(10), 987-991.

The method can avoid the bias introduced by diagenetic effects on the pristine carbon isotopes signature of fossil wood (Barral et al., 2015; Lukens et al., 2019). The data set is analyzed by a relative change in the $\Delta^{13}\text{C}$ value between the time of interest ($t = t$) and a reference time ($t = 0$) ($\Delta(\Delta^{13}\text{C}) = \Delta^{13}\text{C}_{(t)} - \Delta^{13}\text{C}_{(t=0)}$). For example, diagenesis might decrease the $\delta^{13}\text{C}_{\text{wood}}$ values about $\sim 1.4\text{‰}$, which would both increase $\sim 1.4\text{‰}$ in $\Delta^{13}\text{C}_{(t)}$ and $\Delta^{13}\text{C}_{(t=0)}$. However, the $\Delta(\Delta^{13}\text{C})$ do not change, and neither do the pCO_2 estimates. Thus, if the diagenetic effect on the $\delta^{13}\text{C}_p$ is consistent ($\sim 1\text{‰}$), which is likely, the measured $\delta^{13}\text{C}_p$ would still provide consistent CO_2 levels. We have added a discussion on diagenesis and the advantage of this method in the revised manuscript.

There is a large spread in marine carbonate $\delta^{13}\text{C}$ values especially at the peak of the excursion where there is a range of 6‰ . It is hard for me to believe that this degree of uncertainty was fully considered in the error propagation - the CO_2 range shown in Fig 3 seems too narrow. How did you consider the substantial error here? I see the confidence interval on the LOESS fit but it looks way too small, even for a 68% CI. Some of the marine carbonate records show a large excursion (-6‰) whereas others are much smaller. Given that the variable magnitude of marine CIE during the PETM has been attributed to dissolution, would you say that larger marine CIEs give a more accurate representation? The average of the marine CIE may not be the best way to proceed. Please discuss.

Thanks for this comment. We agree that a large variance in marine carbonate $\delta^{13}\text{C}$ values could lead to the large uncertainty in the $\delta^{13}\text{C}$ values of atmospheric CO_2 . Global carbon isotope compilation reveals that marine CIE magnitudes range from -3.0‰ to -4.7‰ ($n = 69$) with median values about -3.8‰ . The variance of marine CIE magnitude is caused by several effects, especially enhanced water stratification and large vertical $\delta^{13}\text{C}$ DIC gradients at deep basin sites during Permian-Triassic crisis, which could cause larger CIE magnitudes (Song et al., 2013). Because of this isotopic gradient, it is unlikely that larger CIE values (e.g. -6‰) represent the true value caused by light carbon release. In addition, the dissolution indeed could decrease the CIE magnitude in deep oceans, but we had stated that the sedimentary facies of our marine sections belong to shallow shelf, deep shelf and slope environments.

Thus, we feel the average value is the best approximation of the global marine CIE magnitude and generates a robust CO_2 estimate. The CIE magnitude in the LOESS fit is about -3.5‰ that is roughly consistent with the median values (-3.8‰). The reliability of LOESS fit had been supported by other two LOESS fittings of subsample and cross-validation method that generates the best degree of smoothing for LOESS fitting. In addition, the errors of $\delta^{13}\text{C}_{\text{carb}}$ are calculated from the LOESS fit of global ten high resolution $\delta^{13}\text{C}_{\text{carb}}$ profile, which range from 0.09‰ to 0.14‰ and produces positive error reaching ~ 150 ppmv and negative error reaching ~ 150 ppmv at $p\text{CO}_{2(t)} = 2000$ ppmv.

How does Fig. 4 support the conclusion that the CIE was not the result of volcanic CO_2 ? I see the black cross overlapping all emission scenarios. Please also discuss whether this calculations assumes rapid emission in which all the carbon goes into the atmosphere, or whether it considers mixing into the ocean. Peak CO_2 will be very different if the release rate is slow and mixing with the ocean occurs.

We had noticed that the black cross also overlaps some parts of volcanic CO_2 results in the previous Fig.4. These parts are estimated from extremely small CIE magnitudes ($1\sim 2\text{‰}$) which are unlikely to represent the global signal. Therefore, this does not show that volcanic CO_2 is main light carbon source. The $p\text{CO}_2$ estimates (black cross) fall mainly within the range of model results for organic matter and methane scenarios.

Whatever, this diagram does cause some confusion.

The carbon isotope mass balance calculation utilizes a carbon reservoir size composed of ocean and atmosphere carbon inventory, but is dominated by the ocean reservoir. Thus, the initial carbon reservoir is assumed to be the initial marine DIC reservoir size (Payne et al., 2010).

Lines 119-120. This is probably right, but to be scientific about it, considering the feedbacks in the carbon cycle, the correlation doesn't mean CO_2 was the control. For instance, methane is highly correlated with CO_2 and temperature during Pleistocene but we don't think of methane as being the primary control.

We agree that the good correlation doesn't mean CO_2 was the only control. The discussion has been modified, emphasizing that CO_2 was likely the dominant greenhouse gas causing global warming, but other greenhouse gases cannot be excluded.

Line 121- where does the 10 degrees C come from? Need global mean T for this calculation.

A compilation of surface sea temperature (SST) from three representative regions (i.e. South China, Iran and Armenia) are showed in Figure 3, which reveals ~10 °C increase in sea surface temperatures. We had added more details about the SST in this sentence.

Line 178-179 - I am not convinced that you can rule out CO₂ > 10,000 ppm using the data presented here. The proxy saturates at high CO₂. Is it really possible to confidently distinguish 5000 from 10,000 ppm?

In fact, the Δ CIE model for evaluating the carbon source is composed of two parts: the hyperbolic relationship between $\Delta^{13}\text{C}$ and $p\text{CO}_2$, and simple carbon isotope mass balance. The former part has large limitation under high CO₂ level. We agree that it is difficult to distinguish different high CO₂ levels by Δ CIE model, which might generate unreliable conclusion. Thus, we simplified the light carbon source method and we used a simpler carbon mass balance to evaluate the carbon source.

The new estimates confirm that the $p\text{CO}_2$ variations calculated from fossil C₃ plants are consistent with modelled carbon emissions from organic matter and methane. There is no overlap between volcanic CO₂ results and estimates from C₃ plant proxy (Fig. 4), which implies that a pure volcanic CO₂ source is not likely. A pure biogenic methane source is also unlikely, because it would result in a too large a CIE (> 6‰). Overall, our new calculations suggest that the volcanic CO₂ alone cannot account for the observed negative CIE, and more ¹³C-depleted sources than volcanic CO₂ are needed to explain the CIE. However, we would like to state that we do not rule out a volcanic CO₂ source, it is likely that multiple sources may have contributed to the total carbon emissions.

List of main revisions of the manuscript according to the comments of Reviewer #2:

1. Text has been added to provide the biological and sedimentological evidence for high and constant precipitation regime during the studied interval.
2. The description of correlation between CO₂ and temperature in the abstract and discussion are corrected.
3. Clarification of diagenesis effect in CO₂ estimate has been provided.
4. A better explanation on the advantage of C₃ plant proxy calculation method has been added.
5. More details on 10 °C increase in temperatures are added.
6. We simplified the method used for the carbon source evaluation and performed carbon isotope mass balance calculation.
7. We revised the discussion of carbon source according to new estimates
8. Figure 4 is replaced based on new estimates.
9. The descriptions of Δ CIE method are changed to carbon isotope mass balance calculation.
10. Parameters for the carbon isotope mass balance calculation Supplementary Table 6 are changed.
11. We deleted the description of sensitivity Analysis of Δ CIE and supplementary figure 8.

Reviewer #3 (Remarks to the Author):

Overall I find this to be a strong manuscript which will be of much interest to a broad audience. The new pCO₂ record represents one of the most high resolution records for the P-T interval. The proxy based method used to generate the new pCO₂ record is not without its critics however there is now ample experimental data which highlight the promise and strengths (and also limitation) of the new C₃ C isotope based proxy. The application of the proxy in this instance is appropriate and robust.

Thanks for your very positive comments.

I find the CIE attribution analysis less convincing because of the range of possible combinations of both CO₂ estimates and ¹³C values for each time interval/ section / possible source. I recommend that the authors down play the certainty of their attribution a little more.

We agree this part has some limitations. We modified the model analysis accordingly and the discussion (see also reply to comments of Reviewer #2).

I have two minor issues with the CO₂ record as presented.

(1) The authors argue that a mixed sample of cuticle, charcoal and various unknown taxonomic groups decrease the likelihood that an evolutionary or phylogenetic signal in ¹³C will interfere or bias the CO₂ record. This is incorrect. An unknown mix of different taxonomic groups means that the potential phylogenetic bias cannot be quantitatively assessed. I agree that an assemblage approach is better than a single species approach (Porter et al) however the authors must add the caveat that any phylogenetic bias in the CO₂ record still remains to be addressed with further study. (lines 487-488 149-150)

We agree with Reviewer #3 that potential phylogenetic bias cannot be ruled out totally due to the mix of different taxonomic groups of plants. The mixture of different plant tissues (i.e. cuticle, charred wood and non-charred wood) used in this study would cause less bias than a single species approach (Porter et al., 2019). We have added this text to clarify: “the mixture of different plant tissues (i.e. cuticle, charred wood and non-charred wood) are used in this study, which very likely include different plant taxa.

(2) The authors do not consider the influence of different chemical treatment on the fidelity of the original ¹³C C₃ plant signal. Baral et al (2015) showed that HF and HCL treated fossil plant material can alter the original ¹³C signal. I note in the methods that different section and different plant preservation types were processed in different ways - some with only HCL, others with both HCL and HF. Were some cuticles also picked without any chemical post treatment prior to ¹³C analysis? I request that the authors address this potential bias in their CO₂ data set. I do not see this as insurmountable. Perhaps it is a consistent bias that can be accounted for? However if different parts of the sections were processed in different ways then this inconsistent bias in the perceived 'raw ¹³C data' and how it may have effected the final CO₂ record must be evaluated.

Barral A, Lécuyer C, Gomez B, Fourel F, Daviero-Gomez V. Effects of chemical preparation protocols on $\delta^{13}\text{C}$ values of plant fossil samples. *Palaeogeography, Palaeoclimatology, Palaeoecology*. 2015 Nov 15;438:267-76.

It is necessary to clarify that the sample treatment method for C_3 plants tissues from four sections are all the same, and involved the use of HCl and HF. The treatment method for bulk organic matter is different from C_3 plants tissues, and involves only HCl. Only carbon isotopes of C_3 plants (see Figs. 3) are used to estimate the $p\text{CO}_2$. Hence, no bias caused by inconsistent methods has been introduced in $p\text{CO}_2$ calculations.

On the other hand, the original $\delta^{13}\text{C}$ signals in C_3 plant might be altered by HCl and HF treatment according to a few experiments (e.g. Barral et al., 2015), and finally bring out potential bias in $p\text{CO}_2$ calculation. This bias can be minimized, because our calculation method can avoid it under specified conditions, even if the carbon isotopes did not preserve original signals. The data set is analyzed by a relative change in the $\Delta^{13}\text{C}$ value between the time of interest ($t = t$) and a reference time ($t = 0$) ($\Delta(\Delta^{13}\text{C}) = \Delta^{13}\text{C}_{(t)} - \Delta^{13}\text{C}_{(t=0)}$). For example, HF might increase the $\delta^{13}\text{C}_p$ values about $\sim 1\text{‰}$, which would both increase $\sim 1\text{‰}$ in $\Delta^{13}\text{C}_{(t)}$ and $\Delta^{13}\text{C}_{(t=0)}$. However, the $\Delta(\Delta^{13}\text{C})$ do not change and so there is no change in CO_2 estimates. Thus, if the bias of $\delta^{13}\text{C}_p$ in all sample caused by HF and HCl is consistent, which is very likely, the non-pristine $\delta^{13}\text{C}$ data would still provide the same CO_2 levels as the original $\delta^{13}\text{C}$ data. We have added the discussion about chemical treatment effect in the revised manuscript. And the description of this method advantage has been added.

References:

- Barral, A., Lécuyer, C., Gomez, B., Fourel, F. and Daviero-Gomez, V., 2015. Effects of chemical preparation protocols on $\delta^{13}\text{C}$ values of plant fossil samples. *Palaeogeography, Palaeoclimatology, Palaeoecology* 438, 267 - 276.
- Feng, Z., Wei, H., Guo, Y., He, X., Sui, Q., Zhou, Y., Liu, H., Gou, X. and Lv, Y., 2020. From rainforest to herbland: New insights into land plant responses to the end-Permian mass extinction. *Earth-Science Reviews* 204, 103153.
- Fielding, C.R., Frank, T.D., McLoughlin, S., Vajda, V., Mays, C., Tevyaw, A.P., Winguth, A., Winguth, C., Nicoll, R.S., Bocking, M. and Crowley, J.L., 2019. Age and pattern of the southern high-latitude continental end-Permian extinction constrained by multiproxy analysis. *Nature Communications* 10, 385.
- Lukens, W.E., Eze, P. and Schubert, B.A., 2019. The effect of diagenesis on carbon isotope values of fossil wood. *Geology* 47, 987-991.
- Payne, J.L., Turchyn, A.V., Paytan, A., DePaolo, D.J., Lehrmann, D.J., Yu, M. and Wei, J., 2010. Calcium isotope constraints on the end-Permian mass extinction. *Proceedings of the National Academy of Sciences* 107, 8543–8548.
- Porter, A.S., Evans-Fitz. Gerald, C., Yiotis, C., Montañez, I.P. and McElwain, J.C., 2019. Testing the accuracy of new paleoatmospheric CO_2 proxies based on plant stable carbon isotopic composition and stomatal traits in a range of simulated paleoatmospheric $\text{O}_2:\text{CO}_2$ ratios.

Geochimica et Cosmochimica Acta 259, 69-90.

Song, H.Y., Tong, J., Algeo, T.J., Horacek, M., Qiu, H., Song, H., Tian, L. and Chen, Z., 2013. Large vertical $\delta^{13}\text{C}$ DIC gradients in Early Triassic seas of the South China craton: implications for oceanographic changes related to Siberian Traps volcanism. *Global and Planetary Change* 105, 7-20.

Yu, J., Broutin, J., Chen, Z., Shi, X., Li, H., Chu, D. and Huang, Q., 2015. Vegetation changeover across the Permian–Triassic Boundary in Southwest China: Extinction, survival, recovery and palaeoclimate: A critical review. *Earth-Science Reviews* 149, 203–224.

REVIEWER COMMENTS

Reviewer #1 (Remarks to the Author):

I have read the revised manuscript, the referees' reports, and the authors' response to reviews. I am satisfied that the authors have addressed all issues raised in my and the other two reviews. I am happy to recommend that the paper be accepted for publication.

Chris Fielding, Lincoln, NE, USA, October 4th, 2020

Reviewer #2 (Remarks to the Author):

The authors have done a good job of addressing some of my concerns and a rather poor job of addressing others. My concerns about rainfall and diagenetic effects have been addressed (although the authors state that the sedimentology suggest no significant change in precipitation and this argument is vague - what about the sediments are you pointing to here?). My concerns remain about the two primary conclusions of the manuscript: 1) the EES estimate and 2) the carbon source.

Earth System Sensitivity requires global mean temperature (unless you are looking at sensitivity in a particular region, but since the authors compare with IPCC, the implication is that this is a global estimate). How do you estimate global temperature from SST in three locations? Do these locations span a range of paleolatitudes? There is no justification given for 10C being the global mean T shift.

How was the mass of carbon added converted to atmospheric CO₂ increase in ppm? (Figure 4 caption). This relates to the point I raised in my first review: "Please also discuss whether this calculations assumes rapid emission in which all the carbon goes into the atmosphere, or whether it considers mixing into the ocean. Peak CO₂ will be very different if the release rate is slow and mixing with the ocean occurs." See Lord et al 2016 Global Biogeochemical Cycles for an analytical expression that might be helpful here. It seems that since the onset of the CIE persist for 75 kyr we are talking about a protracted carbon release, not an instantaneous pulse. The rate of release matters!

Please show the uncertainty of the reconstructed CO₂ increase on figure 4 (the dashed line should be a band) and please describe what the width of that band is based on (percentile values, 2sigmas, etc.)

Finally, and this is more minor, I think clarification on the correlation among marine and terrestrial sections can be reported more clearly. Correlation of marine and terrestrial sections based on d¹³C seems a bit circular given that CO₂ in this paper is calculated from the difference between wood and marine carbonate d¹³C values. Also, even if carbon isotope stratigraphy is a justified means of correlation, the end of the body section of the CIE is not clearly define in some of the wood records. Please discuss and illustrate on a figure the biostratigraphy and other any other tie points used in correlation. This doesn't really influence the conclusion, which are based on the correlations in the onset of the excursion.

Reviewer #3 (Remarks to the Author):

NCOMMS-20-17959A by Y.Y. Wu et al.

Wu et al have provided detailed responses to all of my original queries. Overall, the manuscript is much improved and presents an exciting advance on our understanding of pCO₂ evolution spanning the PTME. I recommend acceptance with minor corrections.

I have three minor remaining comments.

(1) It is not clear what the authors mean by 'diminutive herbaceous vegetation' on Line 165 page 7. Please name the taxa and provide supporting references or a more appropriate ecological classification. Also technically a swamp must have woody plants to be classed as a swamp therefore this should not be referred to as a swamp if it is dominated by herbaceous plants.

(2) Line 200-202. I do not think this statement on carbon sequestration is justified with supporting data. I do not think that the observed prolonged pattern of elevated CO₂ implies a reduced rate of silicate weathering- it may just meant that supply of C far exceeds sequestration. The statement also contradicts the usual assumptions that silicate weathering is a temperature dependent process and positively correlated with temperature.

(3) Figure 3. Please include individual data points for the CO₂ graph to demonstrate the source of data on which the LOWESS curve is based.

We addressed all the reviewers' comments on our manuscript (NCOMMS-20-17959-A). Several of the reviewers suggested changes in wording to the text which we have included in the revised version. Those changes appear as highlights in yellow.

Below are our responses to the reviewers' comments.

REVIEWER COMMENTS

Reviewer #1 (Remarks to the Author):

I have read the revised manuscript, the referees' reports, and the authors' response to reviews. I am satisfied that the authors have addressed all issues raised in my and the other two reviews. I am happy to recommend that the paper be accepted for publication.

Chris Fielding, Lincoln, NE, USA, October 4th, 2020

Thank you for your comments that greatly improved the manuscript.

Reviewer #2 (Remarks to the Author):

The authors have done a good job of addressing some of my concerns and a rather poor job of addressing others. My concerns about rainfall and diagenetic effects have been addressed (although the authors state that the sedimentology suggest no significant change in precipitation and this argument is vague - what about the sediments are you pointing to here?). My concerns remain about the two primary conclusions of the manuscript: 1) the EES estimate and 2) the carbon source.

Thanks for your further comments. We had added the sedimentology of the studied sections in the main text and Supplementary document (Supplementary Figure 8) of the last revision. No significant change in depositional style was noted between the Xuanwei and Kayitou formations, which suggested no significant precipitation change. Especially, there is no evidence for immediate aridification across the mass extinction interval.

Earth System Sensitivity requires global mean temperature (unless you are looking at sensitivity in a particular region, but since the authors compare with IPCC, the implication is that this is a global estimate). How do you estimate global temperature from SST in three locations? Do these locations span a range of paleolatitudes? There is no justification given for 10C being the global mean T shift.

Joachimski et al. (2020) reviewed all SST data calculated based on conodont $\delta^{18}\text{O}$ values during

Permian-Triassic boundary. These data come from three locations (i.e. South China, Iran and Armenia) that all belong to low latitudes, showing the increase of SST ranging from 6.4°C to 11.4°C. The Brachiopods $\delta^{18}\text{O}$ values from middle latitudes (Tibet) also show that >9°C SST increase during this period (Garbelli et al., 2016). Climate model simulations reveal that the increase of SST in high latitude should be higher than low latitude (Penn et al., 2018). However, no SST records in high latitude during Permian-Triassic transition have been reported to support this temperature latitude gradient until now. As a result, the 10 °C increase in sea surface temperature may have been an underestimate of the global SST increase, which leads to an underestimate of the Earth System Sensitivity. In the revision, we added the relevant discussion as follows: “However, climate model simulations reveal that the increase of SST in high latitude should be higher than low latitude (Penn et al., 2018). As a result, the 10°C SST increase in low latitude might underestimate the global SST increase, which leads to an underestimate of the Earth System Sensitivity during the PTME.”

How was the mass of carbon added converted to atmospheric CO₂ increase in ppm? (Figure 4 caption). This relates to the point I raised in my first review: “Please also discuss whether this calculations assumes rapid emission in which all the carbon goes into the atmosphere, or whether it considers mixing into the ocean. Peak CO₂ will be very different if the release rate is slow and mixing with the ocean occurs.” See Lord et al 2016 Global Biogeochemical Cycles for an analytical expression that might be helpful here. It seems that since the onset of the CIE persist for 75 kyr we are talking about a protracted carbon release, not an instantaneous pulse. The rate of release matters!

The current carbon isotope mass balance model to evaluate the carbon source is modified from McNerney and Wing (2011). The cGENIE earth system model shows that 1 Gt C equal to 0.3 ppmv CO₂ (Panchuk et al., 2008). The details of this method are introduced in the method section. We had modified the caption of figure 4.

In simple mass balance calculations, the C emission is usually assumed to be instantaneous (Payne et al., 2010), but the carbon emission during P-Tr is not instantaneous. The $M_{\text{background}}$ represents initial carbon reservoir size including ocean and atmosphere carbon inventory, but dominated by the ocean reservoir. Thus, the $M_{\text{background}}$ is usually assumed to be the initial marine DIC reservoir size (Payne et al., 2010). In addition, mass balance calculations don't consider global carbon cycle processes such as carbon weathering and carbon burial. Overall, there are limitations to this mass balance approach to estimating carbon emissions, while earth system model would be a more powerful tool in the further study. We have modified relevant text that considering the limitations of this method as follows: “However, simple mass balance calculations don't consider global carbon cycle fundamental processes and changes through time, like carbon weathering and burial rates during the studied interval. In addition, the size of the DIC reservoir is usually assumed to be the size of the background surface carbon reservoir, because of poor understanding of atmosphere carbon reservoir size (Payne et al., 2010). These limitations might lead to an underestimate of the

total mass of added CO₂".

Please show the uncertainty of the reconstructed CO₂ increase on figure 4 (the dashed line should be a band) and please describe what the width of that band is based on (percentile values, 2sigmas, etc.)

We have recalculated the increase of $p\text{CO}_2$ (2081 +4764/-1193 ppmv), and added the uncertainty into figure 4. Some relevant texts in caption are also modified: "Second y-axis converts the mass of added carbon to an increase in atmospheric $p\text{CO}_2$ based on the cGENIE earth system model (1 Gt C = 0.3 ppmv CO₂). Grey shaded area represents the 68% confidence intervals of $p\text{CO}_2$ increase (2081 +4764/-1193 ppmv) estimated from the C₃ plant proxy".

Finally, and this is more minor, I think clarification on the correlation among marine and terrestrial sections can be reported more clearly. Correlation of marine and terrestrial sections based on $\delta^{13}\text{C}$ seems a bit circular given that CO₂ in this paper is calculated from the difference between wood and marine carbonate $\delta^{13}\text{C}$ values. Also, even if carbon isotope stratigraphy is a justified means of correlation, the end of the body section of the CIE is not clearly defined in some of the wood records. Please discuss and illustrate on a figure the biostratigraphy and other any other tie points used in correlation. This doesn't really influence the conclusion, which are based on the correlations in the onset of the excursion.

The correlation between marine and terrestrial sections is based on marine and terrestrial carbon isotope profiles that are divided into four stages, which is supported by biostratigraphy (flora, conchostracans) and chronostratigraphy. We make a summary of the flora, conchostracans and chronostratigraphy data from the studied sections, which is shown in a new figure (Supplementary figure 6). The relevant discussions are modified as follows: "the correlation being supported also by biostratigraphy (flora and conchostracans), and radioisotope dating (Supplementary Fig. 6; Supplementary Information)". More details are listed in the supplementary information of geological setting and age model.

Reviewer #3 (Remarks to the Author):

NCOMMS-20-17959A by Y.Y. Wu et al.

Wu et al have provided detailed responses to all of my original queries. Overall, the manuscript is much improved and presents an exciting advance on our understanding of $p\text{CO}_2$ evolution spanning the PTME. I recommend acceptance with minor corrections.

I have three minor remaining comments.

(1) It is not clear what the authors mean by 'diminutive herbaceous vegetation' on Line 165 page 7. Please name the taxa and provide supporting references or a more appropriate ecological classification. Also technically a swamp must have woody plants to be classed as a swamp therefore this should not be referred to as a swamp if it is dominated by herbaceous plants.

Thanks. We have rephrased the text: “The plant community changed from *Gigantopteris* flora-dominated rainforest ecosystem to isoetalean-dominated (lycophyte) herbaceous vegetation that inhabited the surrounding margins of coastal oligotrophic lakes, which indicate fairly constant precipitation regimes during the PTME interval (Yu et al., 2015; Feng et al., 2020)”.

(2) Line 200-202. I do not think this statement on carbon sequestration is justified with supporting data. I do not think that the observed prolonged pattern of elevated CO₂ implies a reduced rate of silicate weathering- it may just mean that supply of C far exceeds sequestration. The statement also contradicts the usual assumptions that silicate weathering is a temperature dependent process and positively correlated with temperature.

We agree with Reviewer #3 that the prolonged pattern of elevated CO₂ means that the rate of carbon supply to the atmosphere (through volcanic degassing, oxidation of organic matter or reduced organic carbon burial, and reduced silicate weathering rates) exceeds the rate of carbon sequestration (carbonate and organic carbon burial). We acknowledge that the silicate weathering rate is dependent on the temperature, but also heavily depends on how fast uplift transports fresh silicate minerals to the Earth surface (Foley, 2015; West, 2012). To clarify the cause of the prolonged pattern of elevated CO₂, we revised the text as the following: “This lengthy phase of extreme warmth likely implies prolonged carbon emissions into ocean-atmosphere system from continued eruption of the Siberian Traps volcanism, and/or reduced carbon sequestration rate, potentially due to lower consumption of atmospheric CO₂ through reduced organic carbon burial (Berner, 2005) and the possible failure of the silicate weathering thermostat (Kump, 2018)”.

(3) Figure 3. Please include individual data points for the CO₂ graph to demonstrate the source of data on which the LOWESS curve is based.

The $p\text{CO}_2$ curve is based on median value, 16th percentile and 84th percentile of calculation results rather than a LOESS curve. The data of dark grey line for $p\text{CO}_2$ are the median values of the 10,000 re-samplings determined by Monte Carlo uncertainty propagation. The upper limit and lower limit of light grey shaded area of $p\text{CO}_2$ are the 16th and 84th percent of 10,000 re-samplings, respectively. Thus, no individual data point was shown in the figure. All the individual data points

are listed in supplementary data.

- Berner, R.A., 2005. The carbon and sulfur cycles and atmospheric oxygen from middle Permian to middle Triassic. *Geochimica et Cosmochimica Acta* 69, 3211-3217.
- Feng, Z., Wei, H., Guo, Y., He, X., Sui, Q., Zhou, Y., Liu, H., Gou, X. and Lv, Y., 2020. From rainforest to herbland: New insights into land plant responses to the end-Permian mass extinction. *Earth-Science Reviews* 204, 103153.
- Foley, B.J., 2015. The role of plate tectonic–climate coupling and exposed land area in the development of habitable climates on rocky planets. *The Astrophysical Journal* 812, 36.
- Kump, L.R., 2018. Prolonged Late Permian–Early Triassic hyperthermal: failure of climate regulation? *Philosophical Transactions of the Royal Society A: Mathematical, Physical and Engineering Sciences* 376, 20170078.
- West, A.J., 2012. Thickness of the chemical weathering zone and implications for erosional and climatic drivers of weathering and for carbon-cycle feedbacks. *Geology* 40, 811-814.
- Payne, J.L., Turchyn, A.V., Paytan, A., DePaolo, D.J., Lehrmann, D.J., Yu, M. and Wei, J., 2010. Calcium isotope constraints on the end-Permian mass extinction. *Proceedings of the National Academy of Sciences* 107, 8543-8548.
- Penn, J.L., Deutsch, C., Payne, J.L. and Sperling, E.A., 2018. Temperature-dependent hypoxia explains biogeography and severity of end-Permian marine mass extinction. *Science* 362, eaat1327.
- Yu, J., Broutin, J., Chen, Z., Shi, X., Li, H., Chu, D. and Huang, Q., 2015. Vegetation changeover across the Permian–Triassic Boundary in Southwest China: Extinction, survival, recovery and palaeoclimate: A critical review. *Earth-Science Reviews* 149, 203–224.

REVIEWER COMMENTS

Reviewer #2 (Remarks to the Author):

I have read this latest version of the manuscript by Wu et al

In my opinion, describing the estimate of ESS as a minimum is an adequate solution to the lack of high latitude temperature data.

I looked at Cui et al 2013 (compare figures 4b and 5a) who used cGENIE to simulate the P-T carbon cycle perturbation. It looks to me like in their simulation, a 20,000 GtC release resulted in an atmospheric CO₂ increase of about 5000 ppm. This (0.25ppm/GtC) is reasonably close to the 0.3ppm/GtC factor used in the current paper and although the Cui et al value makes a volcanic CO₂ source just a little more likely, I am now willing to drop this particular issue. The authors may want to reference Cui et al 2013 to support this part of their argument.

I therefore recommend publication with the following caveats:

1) the 68% confidence interval on CO₂ shown in Fig 4 is fine (although 95% would be better) as long as the authors state in the text that the C3 plant CO₂ proxy saturates at high CO₂ meaning that their approach does not allow them to determine a max CO₂ level. Therefore, it is difficult to truly discount a volcanic CO₂ source. As it stands now, the conclusion that it was not volcanic CO₂ is stated with too much certainty. That conclusion needs to be softened and qualified unless the authors can make a robust argument for it.

2) include a table that shows all the steps in calculating CO₂ using the C3 plant proxy equations. The table that shows the pCO₂ estimates has columns for age, d13Ccarb, SST, d13Cp, D13C, DD13C and pCO₂. Please include a column for the d13C value of atmospheric CO₂ so that the reader can evaluate each step more easily.

3) please state that the uncertainty used in the error propagation (and shown as the error envelopes in figure 3 and in the table reporting the pCO₂ calculations) is the standard error (i.e. uncertainty on the mean value) and not 1 sigma (the table reporting pCO₂ says 1 sigma). It would be helpful to explain in more detail how these uncertainties were determined. The figure 3 caption says they are 68% confidence intervals. But they certainly don't represent 1 sigma of the data - they must be more closely related to standard errors. A clear description of what the errors actually represent and how they were determined will facilitate the readers evaluation the approach taken for error propagation.

We addressed all the reviewer's comments on our manuscript (NCOMMS-20-17959-B). Reviewer #2 suggested changes in wording to the text which we have included in the revised version. Those changes appear as highlights in yellow.

Below are our responses to the reviewers' comments.

REVIEWER COMMENTS

Reviewer #2 (Remarks to the Author):

I have read this latest version of the manuscript by Wu et al

In my opinion, describing the estimate of ESS as a minimum is an adequate solution to the lack of high latitude temperature data.

Thanks for your positive comments. As we stated in Lines 135-137: "the 10 °C SST increase in low latitude might underestimate the global SST increase, which leads to an underestimate of the Earth system sensitivity during the PTME."

I looked at Cui et al 2013 (compare figures 4b and 5a) who used cGENIE to simulate the P-T carbon cycle perturbation. It looks to me like in their simulation, a 20,000 GtC release resulted in an atmospheric CO₂ increase of about 5000 ppm. This (0.25ppm/GtC) is reasonably close to the 0.3ppm/GtC factor used in the current paper and although the Cui et al value makes a volcanic CO₂ source just a little more likely, I am now willing to drop this particular issue. The authors may want to reference Cui et al 2013 to support this part of their argument.

Thanks. We have cited this reference (Cui et al., 2013) in the revised manuscript to support our argument.

I therefore recommend publication with the following caveats:

1) the 68% confidence interval on CO₂ shown in Fig 4 is fine (although 95% would be better) as long as the authors state in the text that the C₃ plant CO₂ proxy saturates at high CO₂ meaning that their approach does not allow them to determine a max CO₂ level. Therefore, it is difficult to truly discount a volcanic CO₂ source. As it stands now, the conclusion that it was not volcanic CO₂ is stated with too much certainty. That conclusion needs to be softened and qualified unless the authors can make a robust argument for it.

Thanks for your further comments. We softened the conclusion as suggested and we modified the text as follows: "Volcanic CO₂ is unlikely to have triggered alone the increase of *p*CO₂ because it is thought to be too heavy (ca. -6‰), and therefore an unrealistically high amount of volcanic CO₂ is required to reproduce the CIE, much higher than the six-fold *p*CO₂ increase calculated from C₃ plants δ¹³C".

Considering the limitations of mass balance calculation and C₃ plant proxy, we've modified the abstract as following: "Mass balance model suggests that volcanic CO₂ is probably not the only

trigger of the carbon cycle perturbation, and that large quantities of ^{13}C -depleted carbon emission from organic matter and methane were likely required during complex interactions with the Siberian Traps volcanism”.

In addition, we have also added the following text in the Lines 200-203: “However, due to the limitation of the C_3 plant proxy, the uncertainty of $p\text{CO}_2$ is significantly larger at high CO_2 levels (Supplementary Fig. 9). Therefore, volcanic CO_2 source could still have made a contribution to the global carbon cycle perturbation.”

2) include a table that shows all the steps in calculating CO_2 using the C_3 plant proxy equations. The table that shows the $p\text{CO}_2$ estimates has columns for age, $\delta^{13}\text{C}_{\text{carb}}$, SST, $\delta^{13}\text{C}_p$, D_{13}C , DD_{13}C and $p\text{CO}_2$. Please include a column for the $\delta^{13}\text{C}$ value of atmospheric CO_2 so that the reader can evaluate each step more easily.

Thanks. The age, $\delta^{13}\text{C}_{\text{carb}}$, $\delta^{13}\text{C}_p$, SST, $\Delta^{13}\text{C}$, $\Delta(\Delta^{13}\text{C})$ and $p\text{CO}_2$ data are available in the previous version (supplementary data 3). We have added the $\delta^{13}\text{C}_{\text{CO}_2}$ data in supplementary data 3 of this version.

3) please state that the uncertainty used in the error propagation (and shown as the error envelopes in figure 3 and in the table reporting the $p\text{CO}_2$ calculations) is the standard error (i.e. uncertainty on the mean value) and not 1 sigma (the table reporting $p\text{CO}_2$ says 1 sigma). It would be helpful to explain in more detail how these uncertainties were determined. The figure 3 caption says they are 68% confidence intervals. But they certainly don't represent 1 sigma of the data - they must be more closely related to standard errors. A clear description of what the errors actually represent and how they were determined will facilitate the readers evaluation the approach taken for error propagation.

Thanks for your comments. The uncertainties of $\delta^{13}\text{C}_{\text{carb}}$, $\delta^{13}\text{C}_p$ and SST are one standard error calculated from LOESS. In addition, the uncertainties of $\delta^{13}\text{C}_{\text{CO}_2}$, $\Delta^{13}\text{C}$, $\Delta(\Delta^{13}\text{C})$ and $p\text{CO}_2$ are the 16th and 84th percentiles calculated by error propagate with Monte Carlo method. The description of the errors in figure 3 caption, method and supplementary data 3 have been revised to make it clearer.